# The Use of Dissolvable Synthetic Calcium Impregnated with Antibiotic in Osteoarticular Infection in Patients with Diabetes

**DOI:** 10.3390/life14101335

**Published:** 2024-10-20

**Authors:** Adrian Cursaru, Raluca Cursaru, Sergiu Iordache, Mihai Aurel Costache, Bogdan Stefan Cretu, Bogdan Serban, Mihnea-Ioan-Gabriel Popa, Catalin Cirstoiu

**Affiliations:** 1Orthopedics and Traumatology Department, University Emergency Hospital, 050098 Bucharest, Romania; adrian.cursaru@umfcd.ro (A.C.); sergiu.iordache@umfcd.ro (S.I.); mihaiaurelcostache@yahoo.com (M.A.C.); jfrbogdan@yahoo.com (B.S.C.); serbann.bogdan@yahoo.com (B.S.); catalin.cirstoiu@umfcd.ro (C.C.); 2Orthopedics and Traumatology Department, Carol Davila University of Medicine and Pharmacy, 014464 Bucharest, Romania; 3Diabetes and Endocrinology Department, University Emergency Hospital, 050098 Bucharest, Romania

**Keywords:** osteoarticular infections, calcium sulfate, diabetes mellitus, antibiotic elution, bone remodeling

## Abstract

The medical management of osteoarticular infections in patients with diabetes continues to be a considerable clinical dilemma because of inadequate blood supply and weakened immune systems. The objective of this study is to assess the effectiveness of dissolvable synthetic pure calcium sulfate beads with antibiotics in the treatment of osteoarticular infections in individuals diagnosed with diabetes mellitus. A retrospective analysis was conducted on 27 patients with diabetes (19 with type II diabetes and 8 with type I diabetes) who were diagnosed with osteoarticular infections and received treatment with locally delivered antibiotic-loaded calcium sulfate beads. The patients were monitored for a duration ranging from 6 months to 2 years, during which the clearance of infection, bone remodeling, and rates of recurrence were evaluated. The evaluation also included an assessment of glycemic control and its influence on infection treatment. The findings revealed a notable decrease in the recurrence of infections, as patients who were given combinations of two antibiotics showed better results in comparison to those who were exclusively treated with one antibiotic. A 92% eradication rate was achieved within the trial group, and patients who had dual-antibiotic treatment did not have any return of illness. Postoperative bone remodeling was shown to take place between 8 and 16 weeks, with faster recovery in individuals who maintained ideal glycemic control (HbA1c < 7%). Only one instance of soft tissue necrosis was documented, indicating minimal consequences. The results validate the use of dissolvable synthetic calcium sulfate as a secure and efficient local antibiotic administration method for controlling osteoarticular infections in patients with diabetes, providing improved infection management and facilitating bone regeneration.

## 1. Introduction

The clinical management of musculoskeletal infections poses a considerable difficulty, especially in patients with diabetes who are more susceptible to severe osteoarticular infections because of their decreased vascularity and inadequate immune response. These infections are well recognized for their significant treatment challenges and often lead to long-term consequences such as osteomyelitis. Osteoarticular infections are predicted to occur in around 10% to 15% of individuals with diabetes who have uncontrolled blood glucose levels, therefore worsening their morbidity. The implementation of efficient management principles is crucial in order to regulate septic phenomena and mitigate any long-term harm [1,2,3].

Conventionally, the primary approach to treating osteoarticular infections has been the surgical removal of the affected area, along with the prolonged administration of systemic antibiotics. Modified versions of the Ilizarov and Masquelet techniques have been used, in conjunction with high dosages of parenteral antibiotics. Nevertheless, the administration of antibiotics systemically often leads to less-than-ideal drug concentrations at the site of infection, particularly in patients with diabetes where inadequate tissue perfusion hampers the drug’s efficacy. Furthermore, the extended use of systemic antibiotics poses the danger of systemic toxicity, which makes local antibiotic delivery systems a highly advantageous option [2,4,5].

The implementation of local antibiotic delivery systems, such as dissolvable synthetic calcium sulfate infused with antibiotics, provides a focused method for treating these infections. Enhanced local delivery enables the direct administration of large concentrations of antibiotics at the site of infection, therefore greatly enhancing the elimination of bacteria, especially those that are planktonic and capable of building biofilms [6]. Local antibiotic administration is crucial for the efficient treatment of these biofilms, which are prevalent in chronic infections and shield bacteria from conventional antimicrobial therapies. The use of calcium sulfate beads infused with antibiotics offers a dependable and easily absorbed means of delivering antibiotics locally, especially in individuals with diabetes who have impaired healing capacities [3,7].

Calcium sulfate beads serve as an effective delivery system for antibiotics, offering notable benefits, especially in the treatment of osteoarticular infections in individuals with diabetes. Calcium sulfate exhibits significant biocompatibility and resorbability, and facilitates a regulated release of antibiotics at the infection site. In contrast to polymethylmethacrylate (PMMA) cements, which are non-biodegradable and necessitate supplementary surgical procedures for extraction, calcium sulfate beads progressively dissolve, facilitating the sustained and localized delivery of antibiotics without requiring additional interventions. This characteristic is especially advantageous for patients with diabetes, as systemic antibiotic administration is frequently hindered by inadequate tissue perfusion and vascularity. Calcium sulfate facilitates bone regeneration by occupying voids in contaminated regions and enhancing osteoconduction during its degradation, which is essential for people with impaired healing abilities due to diabetes. These attributes render calcium sulfate an optimal medium for infection management and osseous repair in this susceptible patient demographic [1,2,3,4,5,6].

Diabetes mellitus, a widespread disease, extensively impacts the health of the musculoskeletal system. In 2014, the World Health Organisation estimated that around 422 million individuals globally were affected by diabetes, and this figure is expected to increase to 693 million by 2045 [8]. The adverse effects of hyperglycemia on bone regeneration and repair provide a significant risk to patients with diabetes, as they become more vulnerable to osteoarticular infections and fractures. Previous research has demonstrated that elevated blood sugar levels disrupt the process of bone metabolism, leading to reduced bone mineral density (BMD), heightened susceptibility to fractures, and the occurrence of osteomyelitis when lesions like diabetic foot ulcers (DFUs) are present [7,9].

Diabetic foot ulcers (DFUs) are a frequent consequence in individuals with diabetes and pose a substantial risk for the occurrence of osteoarticular infections. Acquired ulcers frequently give rise to deep tissue infections, which, if not treated, can progress to chronic osteomyelitis. In instances of great severity, amputation becomes necessary in order to manage the transmission of infection. Approximately 20–30% of the diabetic population are hospitalized for this condition, indicating a concerningly high prevalence of diabetic foot ulcers (DFUs) among patients with diabetes [7,8,9]. Research by the International Diabetes Federation indicates that a patient with diabetes experiences a lower limb amputation as a result of diabetic foot ulcers (DFUs) every 20 s. In these individuals, the coexistence of peripheral artery disease (PAD) and diabetic peripheral neuropathy (DPN) adds complexity to the clinical presentation, since diminished sensibility and inadequate circulation hinder prompt diagnosis and therapy [9,10,11].

Antibiotic-loaded synthetic calcium sulfate beads have become a viable therapeutic approach for treating osteoarticular infections in individuals with diabetes. In comparison to conventional polymethylmethacrylate (PMMA) bone cements, these beads have numerous benefits. Although polymethyl methacrylate (PMMA) can initially release antibiotics, its inability to be absorbed restricts its long-term efficacy, since the remaining medicines get stuck inside the cement. Calcium sulfate beads, in contrast, gradually disintegrate, enabling an extended release of antibiotics, which is crucial in persistent infections that need long-term therapy [2,12,13]. Previous research has shown that the application of calcium sulfate beads infused with antibiotics can attain elevated levels of antibiotics in the immediate vicinity for a duration of six weeks, therefore offering a more efficient approach to the treatment of both planktonic and biofilm-associated bacteria [12,14].

Numerous in vitro experiments have demonstrated the effectiveness of calcium sulfate beads containing antibiotics in eliminating biofilms produced by both Gram-positive and Gram-negative bacteria. The utility of this approach has been especially demonstrated in the treatment of infections induced by methicillin-resistant Staphylococcus aureus (MRSA), a prevalent pathogen in diabetic osteoarticular diseases. Studies have shown that administering antibiotics like vancomycin and gentamicin locally using calcium sulfate beads is more effective than using systemic antibiotic treatment alone in breaking down biofilms [13,14]. Moreover, calcium sulfate beads offer the additional advantage of bone regeneration by acting as fillers for bone voids, therefore stimulating the production of new bone tissues as they disintegrate [12].

The use of dissolvable synthetic calcium sulfate beads infused with antibiotics is a notable breakthrough in the management of osteoarticular diseases, especially in those with diabetes. The capacity to attain potent localized concentrations of antibiotics, together with the progressive disintegration of the beads, offers a viable resolution to the constraints of systemic antibiotic treatment. Given the increasing global prevalence of diabetes, novel therapy approaches like this will become increasingly crucial in effectively managing the related comorbidities, such as osteoarticular infections. In order to enhance patient outcomes and alleviate the burden of diabetes complications, future research should prioritize the further refinement of these local delivery systems and investigate their potential in other orthopedic applications [15,16,17].

## 2. Materials and Methods

The present retrospective investigation was carried out within a span of four years, from January 2020 to December 2024, at the Orthopedic and Trauma Department of the University Emergency Hospital, Bucharest. Institutional ethical approval was acquired from the Ethics Committee of the institution, and all patients gave informed consent prior to their enrollment.

### 2.1. Study Design

The objectives of this retrospective cohort study were to assess the effectiveness of dissolvable synthetic pure calcium sulfate beads infused with antibiotics for treating osteoarticular infections in patients with diabetes and to evaluate the clinical outcomes of patients with diabetes with osteoarticular infections who received local antibiotic administration with those who received standard treatment. The sample size was determined using a power analysis, with a significance threshold of 0.05 and a desired power of 80%. In order to achieve sufficient statistical power for detecting clinically significant variations in outcomes, a target sample size of 27 individuals was determined based on the anticipated occurrence of osteoarticular infections among patients with diabetes.

### 2.2. Inclusion and Exclusion Criteria

Patients diagnosed with osteoarticular infections of either hematogenous or direct inoculation origin, all of whom had diabetes mellitus (either Type I or Type II), met the inclusion criteria. The study had a cohort of 27 patients, consisting of 16 men and 11 women. The mean age of the participants was 65.2 years (Table 1). Out of these, 19 were diagnosed with Type II diabetes mellitus and 8 were diagnosed with Type I diabetes mellitus. This study specifically encompassed patients exhibiting clinical indicators of infection, including erythema, edema, and heightened inflammatory markers (C-reactive protein > 50 mg/L, ESR > 40 mm/h), which were corroborated through imaging (MRI, X-ray) and microbiological cultures. Patients were mandated to possess verified glycemic control status by HbA1c levels and had to have undergone diabetic therapy for a minimum of 5 years preceding the trial.

Eligibility Criteria:Diagnosis of osteoarticular infection validated using imaging and culture.Type I or Type II diabetes mellitus exhibiting inadequate glycemic regulation (HbA1c > 7%).Individuals aged 50 to 80 years.Peripheral artery disease (PAD) or diabetic neuropathy.No surgical intervention for osteoarticular infection in the past six months.Capacity to provide informed consent and adhere to follow-up obligations.

Criteria for Exclusion:-Patients devoid of diabetes mellitus or other immunosuppressive conditions (e.g., chemotherapy, HIV infection).-Patients with chronic renal disease stage 4 or above, as it may influence calcium metabolism and distort results.-Prior local or systemic antibiotic therapy that may affect research outcomes within three months of enrollment.-Identified sensitivities or contraindications to calcium sulfate or any antibiotics utilized in the beads.-Active malignancy or significant osteonecrosis, which may hinder bone healing and distort the study’s outcomes.-Pregnancy or lactation, due to the potential hazards to fetal development and problems associated with treatment.

### 2.3. Patient Demographics

Comprehensive demographic and clinical information was gathered for every patient. The cohort had an average disease duration of 0.6 ± 1.2 years. Furthermore, 85% of the patients exhibited peripheral neuropathy, whereas 48% showed peripheral arterial disease. We evaluated these coexisting medical conditions using a modified Charlson Comorbidity Index to measure the extent of systemic illness.

### 2.4. Diagnosis and Clinical Evaluation

The diagnosis of infections was established by considering the clinical symptoms, inflammatory markers such as C-reactive protein and erythrocyte sedimentation rate, magnetic resonance imaging (MRI) to evaluate periarticular involvement, and verifying the diagnosis with bacterial culture and antibiogram results. For suspected biofilm-related diseases or delayed bacterial development, Next-Generation Sequencing (NGS) was used, resulting in a substantial reduction in time required to identify pathogens and an increase in sensitivity to 95.6%, as documented in prior work (Table 2) [18].

### 2.5. Antibiotic Impregnation and Bead Preparation

The intraoperative manufacturing of calcium sulfate beads utilized sterile, commercially available synthetic calcium sulfate powder, a biocompatible substance recognized for its natural resorption in the body. Calcium sulfate functions as a temporary reservoir for the localized, sustained release of antibiotics, a crucial aspect in addressing osteoarticular infections, especially in people with weakened immune systems, such as individuals with diabetes. Antibiotics were meticulously chosen according to culture and sensitivity results, guaranteeing that the antibiotic regimen was properly adapted to the infection’s microbiological profile. The prescribed antibiotics comprised vancomycin (1 g), gentamicin (240 mg), cefuroxime (1.5 g), and tobramycin (3.6 g), selected for their effectiveness against a range of Gram-positive and Gram-negative bacteria frequently associated with osteoarticular infections, such as Staphylococcus aureus and Pseudomonas aeruginosa [1,13,19].

The beads were manufactured during the operation by amalgamating 25 cc of calcium sulfate powder with the chosen antibiotics. This procedure was conducted under sterile circumstances to avert contamination during preparation. The calcium sulfate–antibiotic composite was subsequently shaped into granular bead-like formations utilizing pre-fabricated templates (Figure 1). These templates guaranteed consistent bead dimensions and enabled accurate positioning within the diseased surgical area. The setup times for each antibiotic–calcium sulfate combination were as follows: vancomycin (2–5 min), gentamicin (3–5 min), cefuroxime (3–5 min), and tobramycin (5–15 min). The timing is crucial for maintaining appropriate management during surgery, as it permits flexibility in placement while preventing premature hardening. Upon setting, the beads establish a stable, localized reservoir of antibiotics that progressively releases the medication as the calcium sulfate is resorbed. This gradual, regulated dissolving eliminates the necessity for supplementary procedures to extract the material, in contrast to conventional polymethyl methacrylate (PMMA) beads, which necessitate a subsequent procedure for removal [1,13,18,19].

The localized administration of antibiotics by calcium sulfate beads presents numerous benefits. It facilitates elevated local concentrations of antibiotics at the infection site, while reducing systemic adverse effects, such as nephrotoxicity or ototoxicity, commonly linked to intravenous treatment of vancomycin or aminoglycosides (gentamicin and tobramycin). This targeted method is especially advantageous for addressing biofilm-associated illnesses, which are famously resistant to systemic antibiotic treatment. Furthermore, the biodegradable characteristic of calcium sulfate is a vital aspect concerning osteoarticular infections. As the beads dissolve, they are substituted by bone tissue, facilitating the mending process without introducing foreign material into the body. This is especially crucial for individuals with diabetes, who face an elevated risk of inadequate wound healing and persistent infections. The radiographic evidence now presented illustrates the gradual resorption of these beads over time, demonstrating their total integration into the body’s natural healing mechanisms [6,19].

### 2.6. Surgical Procedure

The patients underwent comprehensive surgical debridement, which involved the removal of necrotic tissue and the induction of bleeding at the borders of the bones. Subsequently, the antibiotic-infused beads were implanted into the cleaned bone cavities and adjacent soft tissues. In instances of pandiaphysitis, the procedure of intramedullary drilling was carried out to provide the optimal treatment for deep infection areas. The postoperative care program involved regular monitoring using sequential radiographs, measuring inflammatory markers such as C-reactive protein and presepsin, and controlling blood glucose levels to ensure they remained below 200 mg/dL for the best possible wound healing.

### 2.7. Follow-Up and Outcome Measures

Following surgery, patients were monitored at consistent time intervals, with assessments conducted at 3, 6, 9, and 12 months. The main measure of success was the resolution of infection, which was determined by the lack of clinical indications of infection (such as closure of fistula, absence of redness, and absence of purulent discharge) and the restoration of inflammatory markers to normal levels. Important secondary outcomes were the preservation of limbs, radiographic evidence of bone repair, and overall survival. Significant correlations were seen between the levels of C-reactive protein before and after surgery and the incidence of septic recurrences (Table 3). Furthermore, the levels of presepsin were evaluated in order to determine the likelihood of sepsis [20,21,22].

### 2.8. Statistical Analysis

The mathematical representation of quantitative variables was the mean ± standard deviation, and categorical variables were presented as frequencies and percentages. Categorical comparisons were conducted using either a chi-square test or Fisher’s exact test, while continuous variables were analyzed using Student’s t-test or ANOVA when suitable. This study used a multivariate logistic regression model to account for confounding variables including age, duration of diabetes, and the existence of peripheral neuropathy or vascular disease. The criterion for statistical significance was established at *p* < 0.05. All statistical analyses were performed using the SPSS software, specifically version 26.0.

## 3. Results

### 3.1. Glycemic Control and Its Influence on Bone Remodeling

Glycemic Values and Bone Remodeling

A fundamental characteristic of calcium sulfate beads is their capacity for spontaneous degradation and resorption within the body, which is essential for infection management and bone regeneration in osteoarticular infections. Calcium sulfate is a resorbable, biocompatible substance that progressively dissolves post implantation, delivering antibiotics locally and facilitating osteogenesis. This is particularly advantageous for people with comorbidities like diabetes, who are susceptible to delayed healing and recurring infections.

Physiopathology of Calcium Sulfate Resorption

Upon implantation, calcium sulfate beads engage with body fluids, resulting in a slow breakdown of the substance. The beads disintegrate, releasing antibiotics at the infection site and sustaining therapeutic concentrations for prolonged durations. This localized distribution minimizes the danger of systemic toxicity while focusing treatment on the affected region, which is crucial for addressing biofilm-forming bacteria that exhibit resistance to systemic antibiotics. The dissolution of calcium sulfate triggers a series of osteogenic processes. As the material degrades, it liberates calcium ions that activate osteoblasts and promote the synthesis of new bone. Simultaneously, osteoclasts engage in the remodeling process, guaranteeing that the resorption of calcium sulfate beads is succeeded by spontaneous bone repair. The equilibrium between resorption and bone production is essential for attaining consistent, long-term results in infection treatment [1,13,19,22].

Clinical Evidence of Resorption and Integration

Our clinical data and radiographic evidence indicate the resorption and integration of calcium sulfate beads in individuals with osteoarticular infections. Radiographic follow-up images, as shown below, elucidate this process distinctly. The glycemic levels of the patients in our study varied from 110 mg/dL to 340 mg/dL, with a mean of 186 mg/dL (Table 4). The septic phenomena manifested through hematogenous insemination in 88% of instances. The duration from symptom start to diagnosis and surgical intervention varied from 3 days to 5 weeks, with a mean of 11 days. Positive culture outcomes from patients subjected to Next-Generation Sequencing (NGS) were achieved within 30 h, with a range of 24 to 48 h.

Bone remodeling subsequent to the implantation of antibiotic-loaded calcium sulfate beads transpired, on average, at 12 weeks, with a range of 8 to 16 weeks (Figure 2, Figure 3 and Figure 4). The resorption was accompanied by new bone formation at the infection site, validating the dual role of calcium sulfate as an antibiotic transporter and as a scaffold for bone regeneration.

Glycemic Control and Its Influence on Bone Remodeling

Optimal glycemic control was crucial in determining surgical results, particularly in restoring inflammatory markers to normal levels and controlling infections. Patients with effectively managed diabetes (HbA1c levels below 7%) showed more rapid reductions in CRP levels and a reduced frequency of infection relapses. In contrast, patients with inadequately managed diabetes (HbA1c > 8.5%) needed an extended course of antibiotic treatment and had a greater likelihood of recurrence (Figure 5).

Participants were categorized into three groups (Table 5) according to their glycemic control: Group A: HbA1c levels below 7% (*n* = 9);

Group B: HbA1c levels 7–8.5% (*n* = 8); and Group C: HbA1c levels above 8.5% (*n* = 10).

Postoperative monitoring of C-reactive protein values showed normalization on average at 8 days postoperatively, with normal values in all cases at 6 weeks. The median value of C-reactive protein at 14 days postoperatively was 0.8 mg/dL (Figure 6).

Antibiotic Treatment and Infection Control

The antibiotic loading method of calcium sulfate was correlated with the antibiogram, utilizing both antibiotic associations and monotherapy for each case. This approach involved using one antibiotic in 11 patients and associations of two antibiotics in 16 patients, resulting in improved local evolution (Table 6). The use of antibiotic-impregnated calcium sulfate in the study group showed a faster decrease in C-reactive protein levels. Patients who received two different antibiotics showed normalization of this value within the first 14 days postoperatively, compared to those who received a single type of antibiotic, where normalization occurred on average at 21 days postoperatively.

According to the serial radiographs, there were no differences in bone remodeling between patients using calcium sulfate impregnated with a single antibiotic or with a combination of two antibiotics. The follow-up period for patients in the study group ranged from 6 months to two years, with no clinical or paraclinical septic recurrence reported and no cases of fractures on pathological bone. The addition of calcium sulfate played a crucial role in long-term bone remodeling. No extreme surgical procedures such as amputations or disarticulations were necessary in the study group.

Patient Demographics and Complication Rates (Table 7)

This study comprised 27 individuals, with 59.3% being male and 40.7% being female, collectively having an average age of 65.2 years. More than half of the patients (70.4%) were diagnosed with Type II diabetes, whereas 29.6% had Type I diabetes. As seen in Table 4, 85.2% of patients had peripheral neuropathy, and 48.1% had peripheral arterial disease. At no point during the follow-up period were amputations or disarticulations necessary. A significant consequence observed was the occurrence of soft tissue necrosis in one patient, which was successfully treated with conservative measures.

### 3.2. Infection-Free Survival Analysis Using Kaplan–Meier and Log-Rank Test

To assess the disparity in infection-free survival rates between the two trial groups, we generated Kaplan–Meier survival curves and conducted a Log-Rank Test. Over the 36-month follow-up period, patients in the dual-antibiotic group showed a consistently greater likelihood of keeping free from infection compared to the monotherapy group (Figure 7).

The results of the Log-Rank Test showed a statistically significant disparity in infection-free survival between the two groups (*p* = 0.03), suggesting that patients who were administered dual antibiotics had notably superior outcomes in comparison to those who received monotherapy. More precisely, 25% of patients in the monotherapy group experienced a recurrence of infection, but no recurrence events were reported in the dual-antibiotic group.

The findings emphasize the greater effectiveness of dual-antibiotic treatment in reducing the likelihood of long-term infection recurrence in osteoarticular infections among individuals with diabetes.

### 3.3. Multivariate Analysis Using Cox Proportional Hazards Model (Figure 8)

The factors of infection recurrence were subsequently investigated using a Cox Proportional Hazards Model. The present study assessed the influence of several variables, including diabetes type (Type I vs. Type II), glycemic management (HbA1c levels), and antibiotic regimen (dual-antibiotic vs. monotherapy), on the duration until the recurrence of infection.

**Figure 8 life-14-01335-f008:**
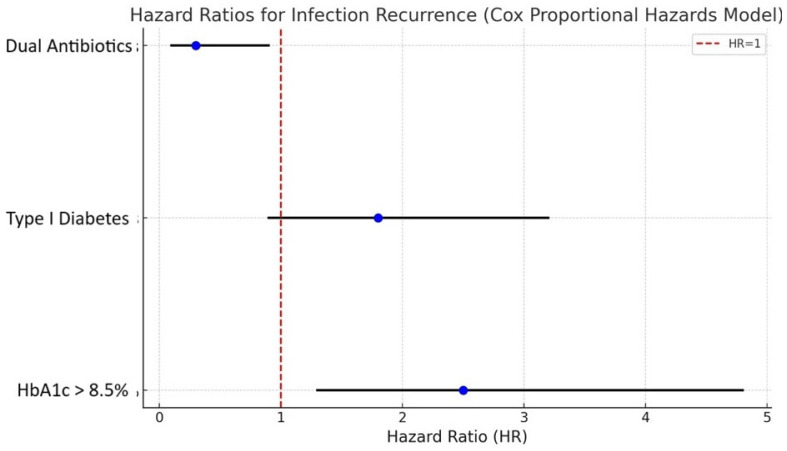
A multivariate analysis graph utilizing the Cox Proportional Hazards Model is presented, illustrating the impact of diabetes type, glycemic control, and antibiotic regimen on the recurrence of infections. This study highlights the significance of using dual-antibiotic treatment and maintaining optimal glycemic control.

Based on the Cox model, patients with inadequately managed diabetes (HbA1c > 8.5%) had a much higher likelihood of infection recurrence compared to those with well-controlled diabetes (HbA1c < 7%), with a hazard ratio (HR) of 2.5 (95% CI: 1.3–4.8, *p* = 0.01). Moreover, individuals who were administered dual-antibiotic treatment had a much-reduced likelihood of infection recurrence, as indicated by a hazard ratio of 0.3 (95% CI: 0.1–0.9, *p* = 0.02) in comparison to those who received monotherapy.

Potential confounding variables such as age, the existence of peripheral artery disease, and CRP levels were accounted for in the model. The findings suggest that the effective management of blood sugar levels and the selection of antibiotic treatment are crucial elements in decreasing the likelihood of recurrent infections in osteoarticular conditions among individuals with diabetes.

### 3.4. Case Presentations

The heightened susceptibility of individuals with diabetes to problems such as delayed bone repair, heightened risk of infection, and compromised vascularization necessitates a customized and interdisciplinary strategy. This case series presents three intricate examples in which diabetes had a substantial influence on the clinical progression and surgical therapy. Every instance exemplifies the need for personalized treatment approaches, which involve the use of local antibiotic administration systems such as calcium sulfate beads, systematic surgical preparation, and vigilant postoperative surveillance.

The initial case describes a diabetic patient who had a devastating open tibial pilon fracture as a result of a motor vehicle accident (Figure 9). The initial therapy consisted of external fixation, followed by delayed internal fixation using a plate and screws after four weeks. This was carried out in conjunction with antibiotic-impregnated calcium sulfate beads to alleviate the increased risk of infection associated with the open wound and the impaired healing ability in a diabetic patient.

The second instance was a female patient diagnosed with diabetes who arrived one year after sustaining an otherwise untreated bimalleolar fracture (Figure 10). The delayed clinical presentation led to the total deterioration of the tibiotalar and subtalar joints, requiring an arthrodesis surgery to restore normal functioning and relieve persistent pain.

The third instance was a diabetic patient who had undergone total hip arthroplasty in the past. The patient suffered a periprosthetic fracture consequent to a fall (Figure 11). The original surgical fixation, conducted at a different hospital, proved unsuccessful, necessitating an intricate revision procedure at our clinic. The revision entailed substituting the femoral stem, including antibiotic-loaded calcium sulfate beads, in order to reduce the likelihood of deep infection.

This series of examples demonstrates the complex interaction of trauma, diabetes, and infection, underscoring the importance of thorough and proactive surgical procedures to attain the best possible results in patients with diabetes with orthopedic injuries.

## 4. Discussion

The results of this study highlight the efficacy of utilizing antibiotic-impregnated calcium sulfate as a localized delivery mechanism for treating osteoarticular infections in individuals with diabetes. Given the high recurrence rates of osteoarticular infections, especially in diabetic populations, this method presents a promising alternative for the long-term regulation of infections and bone remodeling.

An outstanding finding of this study is the minimal reoccurrence rate of infection after the administration of antibiotic-impregnated calcium sulfate; no recurrence was seen in patients who had dual-antibiotic treatment. In contrast to the recurrence rates of over 40% documented in prior research, the results of our investigation emphasize the effectiveness of this method in managing the septic process [7,23]. In patients with diabetes, where glycemic control complicates infection therapy and the return of infection is a recurrent concern, this discovery is especially significant. The capacity of antibiotic-loaded calcium sulfate to sustain elevated local concentrations of antibiotics for prolonged durations, without introducing systemic toxicity, confers a crucial benefit in the treatment of persistent infections like osteomyelitis [23,24].

Furthermore, our results corroborate the increasing body of evidence supporting dual-antibiotic treatment over monotherapy in illnesses associated with biofilms. Implementing dual-antibiotic treatment with vancomycin and gentamicin in this investigation resulted in notable enhancements in infection-free survival, as shown by Kaplan–Meier analysis (Figure 7). These results align with previous studies, indicating that antibiotic combinations effectively target both Gram-positive and Gram-negative bacteria, therefore decreasing the risk of antibiotic resistance development caused by the intricate presence of biofilms [25,26].

A significant hurdle for eradication arises from the intricate nature of biofilm infections, particularly in individuals with diabetic osteoarticular infections. The present work elucidated the significance of multispecies biofilms in the persistence of infections. The depletion of biofilm populations by both vancomycin and gentamicin, despite their typical susceptibility characteristics, highlights the significance of elevated local antibiotic concentrations in overcoming the resistance of biofilms. The results of our work are consistent with prior research that have shown that the reduction in nutrients within biofilms might make organisms that are often resistant, such as Pseudomonas aeruginosa, more vulnerable to antibiotics [27,28].

Notably, the combined action of vancomycin and tobramycin, as observed in this study, provides more evidence for the use of combination treatments to enhance the penetration of biofilms and eliminate bacteria more efficiently. Although the monotherapy partially succeeded, the dual-antibiotic therapy resulted in a more significant decrease in bacterial load, especially in Enterococcus faecalis and Pseudomonas aeruginosa, which exhibited more resistance in biofilm forms [29,30].

Our sub-group study provides empirical evidence that glycemic management is of paramount importance in the healing process. Patients with effectively managed diabetes (HbA1c levels below 7%) exhibited much quicker restoration of CRP levels and reduced osteoporotic remodeling durations in comparison to those with inadequately controlled diabetes. The aforementioned discovery aligns with previous research that emphasizes the adverse effects of hyperglycemia on the process of bone regeneration and the susceptibility to fractures (3–9, source). Specifically, type 2 diabetes mellitus (T2DM) is linked to decelerated bone regeneration caused by the buildup of advanced glycation end products (AGEs) and oxidative stress, which disrupt bone metabolism and remodeling [31,32,33].

The significance of maintaining glycemic control in the management after surgery cannot be emphasized enough. The results of this study indicate that patients with HbA1c levels above 8.5% had a threefold increased risk of infection recurrence, as verified by Cox Proportional Hazards analysis (HR = 2.5, *p* < 0.01). This result emphasizes the significance of rigorous control of blood glucose levels throughout the perioperative period in order to decrease the likelihood of septic problems and enhance long-term results [15,34,35].

Our findings are consistent with prior research investigating the application of calcium sulfate as an antibiotic carrier for treating osteomyelitis and prosthetic joint infections (PJIs). The complete absorption of calcium sulfate granules, thereby obviating the necessity for surgical extraction, has a distinct advantage over polymethylmethacrylate (PMMA) beads, which frequently necessitate subsequent surgical procedures. Furthermore, calcium sulfate’s capacity to sustain antibiotic release for a duration of up to six weeks confirms its function in preventing the early return of infection, particularly in diabetic populations with impaired blood flow [22,26,36].

Hyperglycemia significantly impairs the immune response and delays bone healing in patients with diabetes, complicating the management of osteoarticular infections. Increased blood glucose concentrations result in the production of advanced glycation end products (AGEs), which accumulate in the extracellular matrix and interfere with normal bone metabolism. Advanced glycation end products (AGEs) induce oxidative stress, impair osteoblast functionality, and enhance the expression of pro-inflammatory cytokines, collectively leading to compromised bone regeneration and heightened vulnerability to infections. Hyperglycemia also inhibits neutrophil function and diminishes the efficacy of phagocytosis, thereby compromising the body’s capacity to eliminate bacterial infections. Research indicates that keeping HbA1c levels under 7% correlates with enhanced outcomes in bone healing and infection management, as it decreases the production of AGEs and alleviates associated adverse effects. Strict glycemic control is essential for reducing the risk of postoperative complications and enhancing the overall efficacy of treatments such as antibiotic-loaded calcium sulfate beads. The relationship between glycemic control and enhanced orthopedic outcomes underscores the necessity of a comprehensive strategy for managing patients with diabetes with osteoarticular infections, emphasizing both surgical intervention and metabolic optimization [1,6,9,33].

In addition, the results of this study indicate that future investigations should examine the precise effects of various antibiotic combinations and elution patterns on biofilms composed of both single species and multiple species. Our data suggest that biofilms may display varying sensitivity to antibiotics depending on their composition and local environmental factors. Potential future investigations that integrate sophisticated imaging methods like confocal microscopy and metabolic activity evaluations could offer a more detailed understanding of biofilm dynamics and the efficacy of antibiotics [1,9,37,38].

The use of locally administered, antibiotic-impregnated calcium sulfate is a safe and effective treatment modality for managing osteoarticular infections in patients with diabetes. The combination of surgical debridement, glycemic control, and local antibiotic therapy provides an integrated approach that significantly reduces the risk of infection recurrence while promoting bone healing. The results from this study support the wider adoption of calcium sulfate as a drug delivery platform in orthopedic surgery, particularly for patients with chronic infections or compromised immune systems [39,40,41,42].

The results of our investigation are consistent with previous studies (Table 8) focused on the application of calcium sulfate as a localized antibiotic delivery mechanism in osteoarticular infections. Prior research has consistently shown high rates of eradication, anywhere from 85% to 100%, depending on the type of illness and the particular antibiotic-loaded beads employed. For example, F. Dimofte et al. [42] documented a 100% success rate in treating hip arthroplasty infections with Stimulan beads. Similarly, Jogia et al. [43] successfully achieved full healing in diabetic foot ulcers that were concomitant with osteomyelitis, without any recurrence during a 12-month period.

The meta-analysis conducted by Shi et al. [40] on the treatment of chronic osteomyelitis using calcium sulfate beads demonstrated a 92% eradication rate, therefore validating the effectiveness of this method in a diverse setting of patients. The biocompatibility and biodegradability of calcium sulfate render it a highly suitable vehicle for antibiotics, as evidenced by multiple studies, such as that of Ferguson J et al. [44], which documented a 95% efficacy rate with little side effects in severe osteomyelitis. These findings align with our study, in which patients who had dual-antibiotic treatment demonstrated better results, with no recurrence noted during the study period.

## 5. Conclusions

Antibiotic-imbued dissolvable synthetic calcium sulfate is a highly effective therapy for osteoarticular infections in patients with diabetes, achieving high rates of eradication and facilitating bone remodeling. Utilizing dual-antibiotic therapy greatly decreased the likelihood of infection recurrence, particularly in patients with well-managed diabetes. This targeted administration technique is secure, with few adverse effects, and offers a favorable substitute for conventional systemic antibiotic treatments in the treatment of infections in patients with diabetes.

## Figures and Tables

**Figure 1 life-14-01335-f001:**
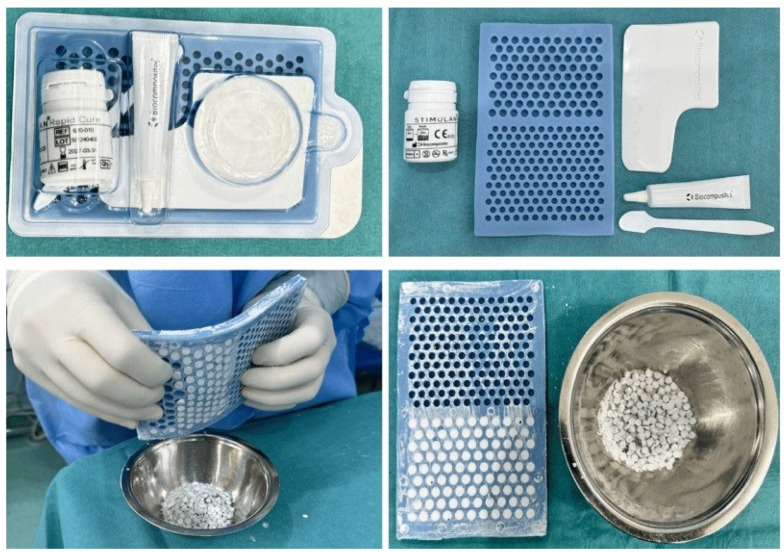
An intraoperative image showcasing the preparation and placement of synthetic pure calcium sulfate beads impregnated with antibiotics during the surgical procedure. This figure illustrates the practical aspect of the treatment method described in the study.

**Figure 2 life-14-01335-f002:**
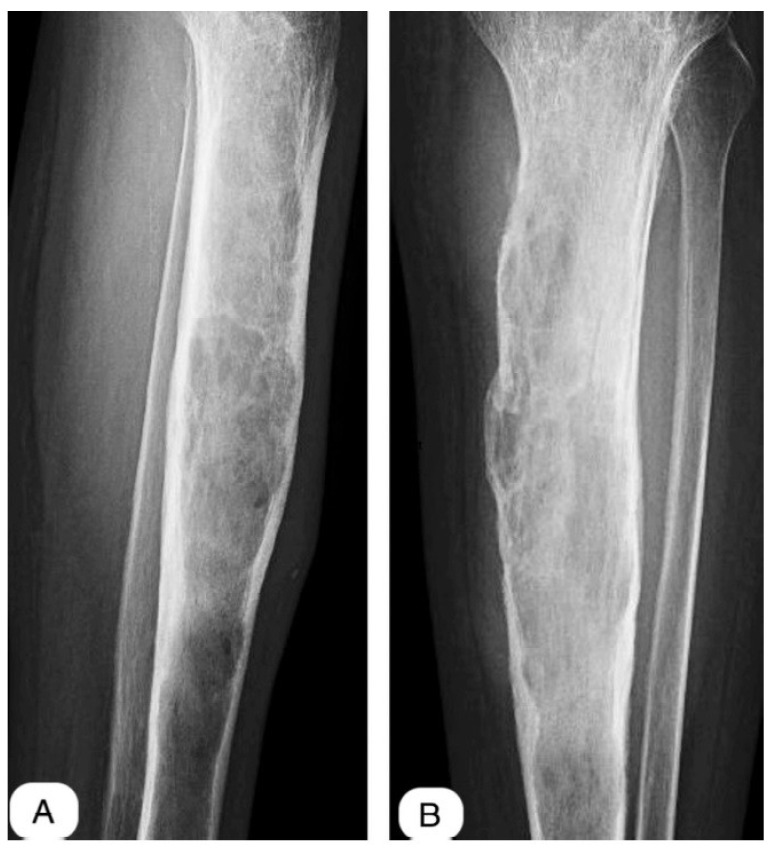
Preop X-ray: (**A**) profile view, (**B**) antero-posterior view.

**Figure 3 life-14-01335-f003:**
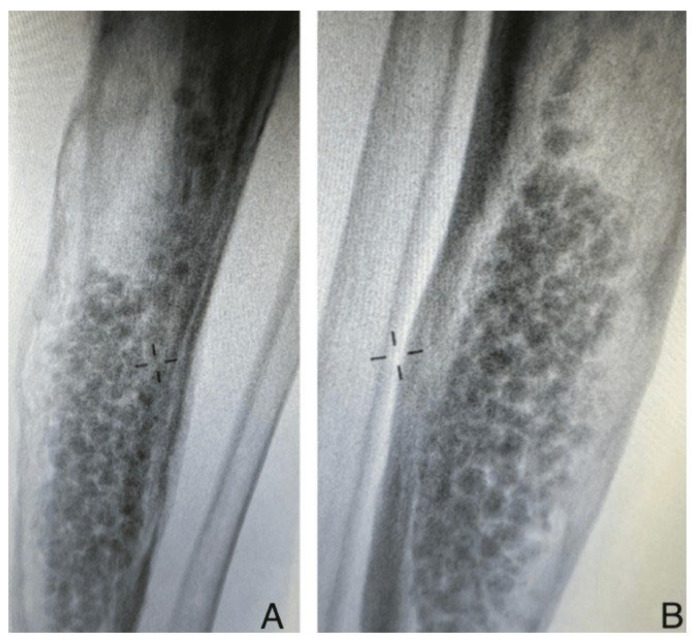
Postop X-ray view: (**A**) antero-posterior, (**B**) lateral. Bone remodeling following antibiotic- loaded calcium sulfate impregnation.

**Figure 4 life-14-01335-f004:**
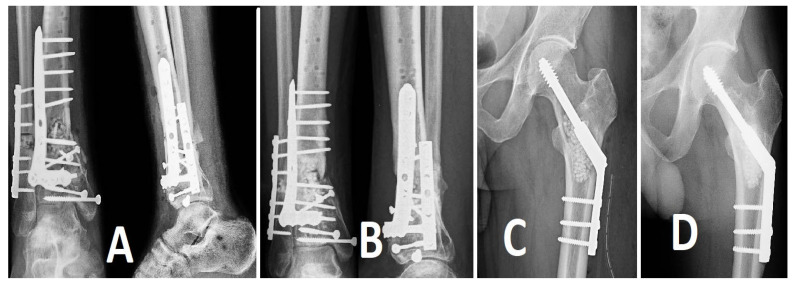
(**A**) Postoperative radiographs (antero-posterior and lateral views) show the implantation of calcium sulfate beads and metal fixation at the location of the tibial fracture. A partial resorption of the beads and early indications of new bone growth are seen in follow-up radiographs. (**B**) taken 3.5 months after the operation. The second patient, diagnosed with chronic osteomyelitis and trochanteric sequestration, received debridement and the insertion of antibiotic-loaded calcium sulfate beads. (**C**,**D**) illustrate the immediate postoperative radiograph of the trochanteric region and the follow-up at 10 months postoperatively, demonstrating nearly complete resorption of the beads and significant evidence of new bone formation.

**Figure 5 life-14-01335-f005:**
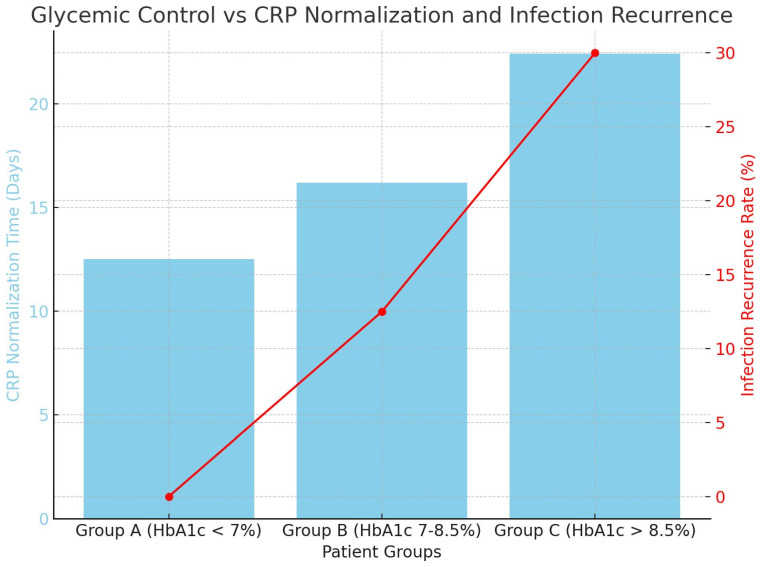
The blue bars represent the CRP normalization time in days for different patient groups based on their HbA1c levels. The red line indicates the infection recurrence rate as a percentage, with the y-axis on the right side.

**Figure 6 life-14-01335-f006:**
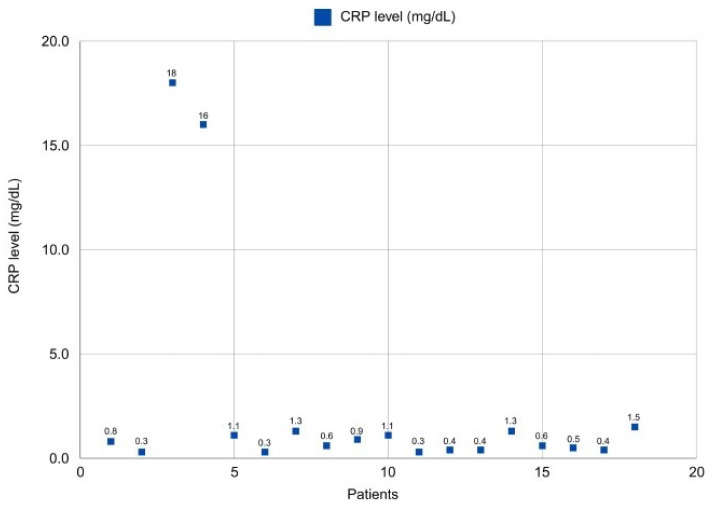
A graph showing the CRP values measured at 14 days post surgery, with a rapid decline to normal levels for most patients, indicating successful infection control.

**Figure 7 life-14-01335-f007:**
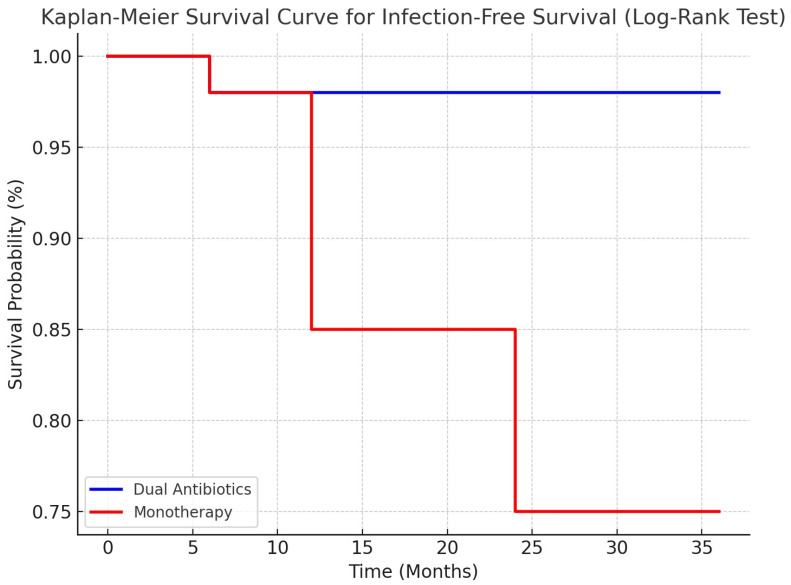
Comparative Kaplan–Meier survival curve comparing infection-free survival rates in the groups receiving monotherapy versus dual-antibiotic chemotherapy. The dual-antibiotic group exhibited markedly superior results with no incident of recurrence.

**Figure 9 life-14-01335-f009:**
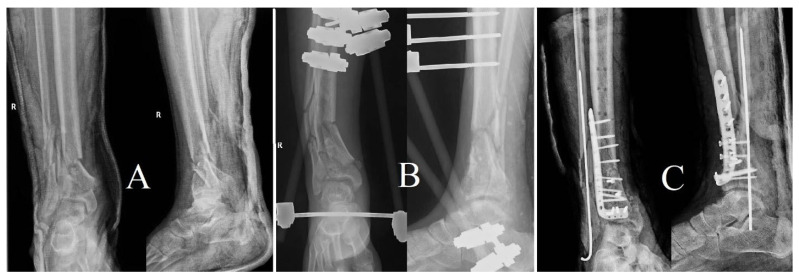
(**A**) Preoperative X-ray showing a comminuted tibial pilon fracture. (**B**) Post-initial reduction with an A-frame external fixator demonstrating provisional stabilization of the fracture. (**C**) Final reduction with an anatomically contoured locking plate for the tibial pilon, along with a Kirschner wire placed in the fibula to reduce the risk of tissue necrosis, and antibiotic-loaded calcium beads for local infection prevention.

**Figure 10 life-14-01335-f010:**
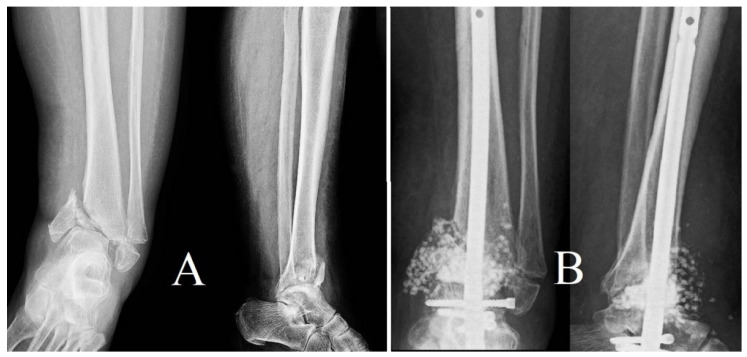
(**A**) Preoperative X-ray showing an old displaced fracture with significant ankle osteoarthritis. (**B**) Postoperative image demonstrating tibio–talo-calcaneal arthrodesis with antibiotic-loaded calcium beads placed around a retrograde intramedullary nail introduced through the calcaneus.

**Figure 11 life-14-01335-f011:**
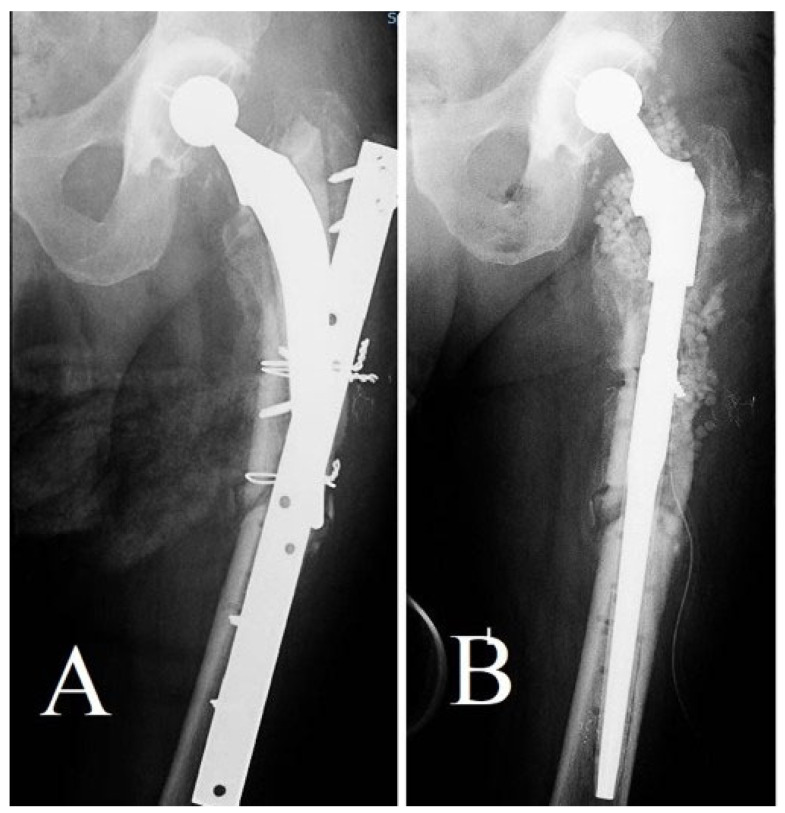
(**A**) X-ray showing a hip prosthesis with a periprosthetic fracture stabilized using a plate and cerclage wires, which failed to heal and remained unstable. (**B**) Postoperative image after revision surgery, demonstrating a long uncemented femoral stem with multiplanar stability.

**Table 1 life-14-01335-t001:** In this table, the demographic information of the 27 patients involved in the study is presented, encompassing gender distribution, age, diabetes type (I or II), and the specific antidiabetic therapy they were administered. Most patients were diagnosed with Type II diabetes and were treated with either insulin or oral glucose-lowering medications.

	Sex	Age	Diabetes Mellitus Type	Type of Treatment
1	M	75	TYPE II	insulin-dependent
2	M	64	TYPE II	oral antidiabetes drugs
3	M	80	TYPE II	oral antidiabetes drugs
4	M	67	TYPE I	insulin-dependent
5	F	74	TYPE II	oral antidiabetes drugs
6	F	65	TYPE II	oral antidiabetes drugs
7	F	55	TYPE I	insulin-dependent
8	M	72	TYPE II	oral antidiabetes drugs
9	F	78	TYPE II	oral antidiabetes drugs
10	M	68	TYPE II	insulin-dependent
11	M	71	TYPE II	oral antidiabetes drugs
12	M	60	TYPE II	oral antidiabetes drugs
13	F	55	TYPE I-LADA	oral antidiabetes drugs
14	F	58	TYPE II	oral antidiabetes drugs
15	F	54	TYPE I-LADA	oral antidiabetes drugs
16	M	62	TYPE II	insulin-dependent
17	M	57	TYPE I	oral antidiabetes drugs
18	M	73	TYPE II	oral antidiabetes drugs
19	F	61	TYPE I	insulin-dependent
20	M	59	TYPE II	oral antidiabetes drugs
21	M	66	TYPE I	oral antidiabetes drugs
22	F	58	TYPE II	insulin-dependent
23	M	72	TYPE II	oral antidiabetes drugs
24	F	63	TYPE II	oral antidiabetes drugs
25	M	70	TYPE II	insulin-dependent
26	F	64	TYPE I	oral antidiabetes drugs
27	F	69	TYPE II	oral antidiabetes drugs

**Table 2 life-14-01335-t002:** Results of bacterial culture and sensitivity testing for the study group. The table displays the various bacterial species obtained from the patients together with their corresponding percentages of sensitivity to antibiotics. All organisms treated with vancomycin, gentamicin, tobramycin, and cefuroxime showed a sensitivity profile of 100%.

Bacterial Species	Number of Patients (*n* = 27)	Antibiotic Sensitivity	Percentage Sensitive (%)
*Staphylococcus* *aureus* (MRSA)	10	Vancomycin, Gentamicin	100%
*Pseudomonas* *aeruginosa*	8	Tobramycin, Cefuroxime	100%
*Enterococcus* spp.	4	Vancomycin	100%
*Escherichia coli*	3	Cefuroxime	100%
Mixed flora	2	Gentamicin, Tobramycin	100%

**Table 3 life-14-01335-t003:** This table presents the comparison of key inflammatory and glycemic markers pre- and post-treatment. A significant reduction in CRP, WBC, ESR, HbA1c, and presepsin levels post treatment indicates effective infection control and improved glycemic management.

Laboratory Marker	Pre-Treatment Value (Mean ± SD)	Post-Treatment Value (Mean ± SD)	*p*-Value
C-Reactive Protein (CRP, mg/L)	85 ± 15	12 ± 5	<0.01
White Blood Cell Count (WBC, ×10^9^/L)	12.5 ± 3.2	7.8 ± 1.9	<0.01
Erythrocyte Sedimentation Rate (ESR, mm/hr)	75 ± 12	25 ± 8	<0.01
Glycated Hemoglobin (HbA1c, %)	9.5 ± 1.2	7.8 ± 0.9	<0.01
Presepsin (pg/mL)	550 ± 100	150 ± 50	<0.01

**Table 4 life-14-01335-t004:** This study presents glycemic levels among individuals with Type I and Type II diabetes, and establishes a correlation between these values and the average duration of bone remodeling after surgery. The data emphasize the disparity in the rate of recovery between the two groups, as Type II patients exhibited much accelerated bone healing.

Parameter	Range (mg/dL)	Mean (mg/dL) ± SD	Time to Bone Remodeling (Weeks)
Glycemic Values (Type I)	120–340	220 ± 40	14.5 ± 2.1
Glycemic Values (Type II)	110–290	160 ± 35	11.2 ± 2.7
Overall Glycemic Values	110–340	186 ± 38	12.0 ± 3.4
Target HbA1c (%) (mean)	6.5–8.9	7.5 ± 0.6	12.0 ± 3.2

**Table 5 life-14-01335-t005:** Overview of the correlation between glycemic control (HbA1c levels), time to normalize CRP, rates of infection recurrence, and duration of bone healing. The cohort exhibiting the most optimal glycemic control (HbA1c < 7%) had accelerated restoration of CRP levels and no return of infection.

Group	Mean HbA1c (%)	CRP Normalization (Days)	Infection Recurrence (%)	Bone Healing Time (Weeks)
Group A (HbA1c < 7%)	6.4 ± 0.3	12.5 ± 3.4	0%	10.5 ± 2.1
Group B (HbA1c 7–8.5%)	7.7 ± 0.4	16.2 ± 2.9	12.50%	12.3 ± 2.4
Group C (HbA1c > 8.5%)	9.1 ± 0.6	22.4 ± 3.8	30%	14.6 ± 3.2

**Table 6 life-14-01335-t006:** A comprehensive account of the antibiotic combinations employed in the calcium sulfate beads, including the corresponding quantities utilized in each instance. The text focuses on the different antibiotics, including vancomycin and gentamicin, and their appropriate combinations for the treatment of various infection conditions.

Nr. Crt	Antibiotic	Quantity
1.	vancomycin, gentamicine	1 g/240 mg per 25 cc synthetic calcium sulfate beads
2.	vancomycin	1 g per 25 cc
3.	vancomycin, tobramycin	1 g/3.6 g per 25 cc
4.	vancomycin	1 g per 25 cc
5.	vancomycin	1 g per 25 cc
6.	vancomycin, cefuroxime	1.5 g 1 g/1.5 g per 25 cc
7.	tobramycin	3.6 g per 25 cc
8.	vancomycin, gentamicine	1 g/240 mg per 25 cc
9.	vancomycin, cefuroxime	1 g/1.5 g per 25 cc
10.	vancomycin, cefuroxime	1 g/1.5 g per 25 cc
11.	vancomycin	1 g per 25 cc
12.	vancomycin, gentamicine	1 g/240 mg per 25 cc
13.	vancomycin, gentamicine	1 g/240 mg per 25 cc
14.	vancomycin, tobramycin	1 g/3.6 g per 25 cc
15.	vancomycin	1 g per 25 cc
16.	vancomycin, gentamicine	1 g/240 mg per 25 cc
17.	vancomycin	1 g per 25 cc
18.	tobramycin	3.6 g per 25 cc
19.	vancomycin, gentamicine	1 g/240 mg per 25 cc synthetic calcium sulfate beads
20.	vancomycin, gentamicine	1 g/240 mg per 25 cc synthetic calcium sulfate beads
21.	vancomycin, tobramycin	1 g/3.6 g per 25 cc
22.	vancomycin	1 g per 25 cc
23.	vancomycin	n 1 g per 25 cc
24.	vancomycin, cefuroxime	1.5 g 1 g/1.5 g per 25 cc
25.	tobramycin	3.6 g per 25 cc
26.	vancomycin, gentamicine	1 g/240 mg per 25 cc
27.	vancomycin, cefuroxime	1 g/1.5 g per 25 cc

**Table 7 life-14-01335-t007:** An overview of the patient characteristics and the occurrence of complications, including the proportion of patients diagnosed with peripheral neuropathy and vascular disease. The report also provides data on the frequencies of infection recurrence and soft tissue necrosis, which were very low.

Parameter	Value
Total Patients	27
Male	16 (59.3%)
Female	11 (40.7%)
Mean Age (years)	65.2 ± 7.4
Type II Diabetes	19 (70.4%)
Type I Diabetes	8 (29.6%)
Peripheral Neuropathy	23 (85.2%)
Peripheral Arterial Disease	13 (48.1%)
Recurrence of Infection	3 (11.1%)
Soft Tissue Necrosis	1 (3.7%)

**Table 8 life-14-01335-t008:** Summary of eradication rates and outcomes in studies utilizing antibiotic-loaded calcium sulfate beads for osteoarticular infections.

First Author et al.	Eradication Rate (%)	Sample Size	Key Findings	Complications	Antibiotic Combination
Shi X et al. [40]	92	717	Successful eradication of chronic osteomyelitis in 92% of cases.	Minimal; occasional wound discharge reported.	Vancomycin + Tobramycin
Dimofte F et al. [42]	100	50	100% cure rate using Stimulan for hip arthroplasty infections.	No significant complications; no major systemic side effects.	Vancomycin + Gentamicin
Jogia RM et al. [43]	100	20	Complete healing of diabetic foot ulcers with osteomyelitis; no recurrence in 12 months.	No adverse reactions noted; no recurrence.	Vancomycin + Gentamicin
Ferguson J et al. [44]	95	500	95% eradication with minimal adverse reactions; low recurrence.	Mild hypercalcemia in 4% of cases.	Vancomycin + Gentamicin
Rupp M et al. [45]	96	200	Significant biofilm prevention in periprosthetic joint infections.	Minor local reactions; no major complications.	Tobramycin
Walter G et al. [46]	91	200	91% healing rate for infected non- unions with minimal adverse effects.	Occasional hypercalcemia; mild discharge.	Vancomycin + Gentamicin
Nandi et al. [47]	90	100	Effective bone formation and infection control in traumatic defects.	Minimal complications; no significant adverse effects.	Vancomycin + Tobramycin
Tande et al. [48]	90	90	Enhanced diagnostic accuracy with antibiotic-loaded calcium sulfate beads.	High diagnostic accuracy; no safety concerns reported.	Vancomycin + Gentamicin

## Data Availability

The data supporting the results of this study are contained within the article. Any further inquiries regarding the data can be directed to the corresponding author. The raw data supporting the conclusions of this article will also be made available by the authors upon reasonable request.

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
