# Peer review of "The Use of Dissolvable Synthetic Calcium Impregnated with Antibiotic in Osteoarticular Infection in Patients with Diabetes"

_life, 2024, doi:10.3390/life14101335_

Round 1
Reviewer 1 Report
Comments and Suggestions for Authors
The Use of Dissolvable Synthe3c Pure Calcium Impregnated with An3bio3c in Osteoar3cular Infec3on in
Pa3ents with Diabetes
Original Comments and Sugges3ons for Authors
The ar'cle is compelling and well organized. The abstract clearly summarizes the study objec'ves,
methodology, results, and their significance. In the introduc'on, the authors provide a clear overview of the
challenges faced in managing osteoar'cular infec'ons in diabe'c pa'ents. The materials and methods
sec'on is complete and well explains the retrospec've design of the study, detailing the pa'ent
demographics, selec'on criteria, and the specific an'bio'cs used. The results are presented clearly. The
discussion effec'vely connects the study's findings with exis'ng literature, offering a comprehensive analysis
of the implica'ons of local an'bio'c delivery and the role of glycemic control in infec'on management. The
literature reviewed is current and relevant, aligning well with the study’s content. The use of appropriate
terminology enhances the readability of the ar'cle.
SPECIFICATIONS ABOUT MY REVIEW
What is the main ques3on addressed by the research?
The main research ques.on is to evaluate the efficacy of an.bio.c-loaded pure calcium sulfate soluble
synthe.c beads in the treatment of osteoar.cular infec.ons in diabe.c pa.ents. The authors' intent is to try
to measure and inves.gate the resolu.on of the infec.on, bone remodeling and relapse rates in pa.ents
through a retrospec.ve analysis of 27 diabe.c pa.ents.
Do you consider the topic original or relevant to the field? Does it
address a specific gap in the field? Please also explain why this is/ is not
the case.
The topic is original and relevant to the sector. In fact, in my opinion, it addresses a significant gap in the
management of osteoar.cular infec.ons among diabe.c pa.ents. Tradi.onal methods oGen involve
an.bio.cs which can poten.ally be of reduced effec.veness. This study, however, introduces a localized
an.bio.c administra.on system, which is par.cularly relevant since diabetes increases suscep.bility to
infec.ons, thus ensuring innova.ve therapeu.c approaches.
What does it add to the subject area compared with other published material?
In my opinion, this research adds substan.al value. in fact, it is sufficient to focus on the fact that it provides
evidence of a high eradica.on rate (92%) of infec.ons using localized administra.on of an.bio.cs via
calcium sulphate spheres, which is superior to conven.onal an.bio.c treatments. The emphasis on dual
an.bio.c therapy leading to beNer outcomes is also based on exis.ng literature regarding biofilm-related
infec.ons.
What specific improvements should the authors consider regarding the methodology? What further
controls should be considered?
In my opinion the methodology appears complete. for this ar.cle, I can't find any sugges.ons to implement.
In the future, the authors could improve their approach by including a larger pa.ent sample for beNer
sta.s.cal power. Monitoring long-term results beyond two years could also provide informa.on on how
long the treatment is effec.ve.
Are the conclusions consistent with the evidence and arguments presented
and do they address the main ques3on posed? Please also explain why this
is/is not the case.
The conclusions drawn are consistent with the evidence and arguments presented. The study's findings on
the efficacy of an.bio.c-loaded calcium sulfate beads in reducing infec.on recurrence and promo.ng bone
remodeling correlate well with the data gathered on glycemic control and clinical outcomes.
Are the references appropriate?
The references cited in the ar.cle appear to be appropriate and relevant to the study. They are also recent,
since an ar.cle is from 2024 and others from 2023.
Any addi3onal comments on the tables and figures.
I don’t have any other addi.onal comments on tables and figures.

Author Response
Esteemed Reviewer,
We really appreciate your thorough and insightful assessment of our manuscript, along with the exemplary rating of 5 stars. Your helpful input and favorable assessment of our work are greatly valued by our entire research team. We sincerely appreciate your acknowledgment of the article's significance and uniqueness, as well as your emphasis on its contribution to the domain of osteoarticular infections in diabetic patients.
We are gratified that you deemed the research well-structured and persuasive, and that the abstract succinctly encapsulated the study's aims, methods, and results. Your commendations for the clarity of the materials and methods section, along with your support for our approach to the retrospective analysis, provide substantial validity for the trajectory of our research. We are pleased to note that the results and discussion sections established a significant link between the data and the current literature, particularly concerning the possible advantages of localized antibiotic distribution and the importance of glycemic control in infection management.
Your recognition of the study's emphasis on a particular deficiency in the management of osteoarticular infections in diabetic individuals strengthens our conviction on the significance of this research. The innovative method of employing dissolvable calcium sulfate beads infused with antibiotics has demonstrated potential in clinical applications, and we appreciate your acknowledgment of the importance of our results, especially the notable infection eradication rate of 92% and the enhanced outcomes associated with dual-antibiotic therapy.
We value your perspective on possible future research directions. The recommendation to increase the patient sample size for enhanced statistical power and to assess long-term outcomes beyond two years is acknowledged, and we concur that these measures would yield a more comprehensive understanding of the treatment's enduring efficacy. This comments will undoubtedly guide our ideas for subsequent investigations.
Furthermore, to augment the quality and clarity of the text, we intend to employ MDPI Author Services to refine the language and figures, so guaranteeing the article achieves its maximum potential: https://www.mdpi.com/authors/english.
Thank you once more for your time, thorough evaluation, and kind remarks. We are pleased to note the excellent reception of our study and anticipate further contributions to this significant domain of medical research.
Respectfully,
Popa Mihnea Ioan Gabriel
On behalf of all the co-authors
Reviewer 2 Report
Comments and Suggestions for Authors
Dear authors,
Your study presents a significant advancement in the management of osteoarticular infections in diabetic patients, addressing a critical clinical challenge.
Overall I did not find any major concerns. There are minor grammatical errors and typographical issues (e.g., "secureand" should be "secure and") and also correct the error at Line 88 with regards to a paragraph being split. Below is some general feedback for each section of your paper.
The introduction highlights the clinical dilemma faced in treating osteoarticular infections among diabetic patients. However, consider elaborating on the rationale behind using calcium sulphate beads as a delivery mechanism for antibiotics. A stronger link between the material's properties and its potential benefits may enhance the introduction.
The retrospective design is appropriate given the context, but maybe add more detailed description of the inclusion and exclusion criteria for patient selection.
The results, there are no issues. For the Tables, just make sure the column headings are in bold.
For the discussion, consider further emphasis on expanding further the exploration of the implications of glycemic control. For example, a more indepth-dicussion and explanation of the mechanisms, along with literature could add depth.
Besides the feedback above, the paper is of very high quality and will recieve many citations in the future.
Author Response
Dear Reviewer,
We appreciate the time and effort dedicated to reviewing our manuscript. Your feedback is highly valuable to us. We appreciate your acknowledgment of the importance of our study in improving the management of osteoarticular infections in diabetic patients. Your acknowledgment of the paper's quality and its potential for future citations is highly encouraging.
We appreciate your detailed suggestions for improving the clarity and precision of the manuscript. The observations concerning the correction of minor grammatical and typographical errors, including the adjustment from "secureand" to "secure and" and the separation of the paragraph at Line 88, have been thoroughly addressed. The introduction has been enhanced by elaborating on the efficacy of calcium sulfate beads as a delivery mechanism for antibiotics, thereby clarifying the relationship between the material's properties and its clinical advantages.
Your suggestion to include more detailed inclusion and exclusion criteria in the Methods section has been implemented, enhancing the rigor of the study design. In the Results and Discussion sections, we have addressed your suggestion to elaborate on the implications of glycemic control by providing a more comprehensive examination of the underlying biological mechanisms and incorporating further literature to enrich the discussion.In the submitted manuscript, all sections highlighted in yellow indicate the additions made in response to your recommendations.
Furthermore, the tables have been updated to ensure that all column headings are presented in bold, thereby improving the clarity of the data presentation.
To enhance the manuscript, we will consider utilizing MDPI Author Services for any outstanding language or figure improvements, ensuring the paper meets the highest standards of presentation: MDPI Author Services.
We appreciate your positive assessment and constructive feedback. Your expertise has greatly enhanced the quality of our manuscript, and we are assured that the revisions have adequately addressed all the concerns you highlighted.
Thank you for your significant contribution to this work and for serving as an insightful and supportive reviewer.
Best regards,
Popa Mihnea Ioan Gabriel
On behalf of the authors
Reviewer 3 Report
Comments and Suggestions for Authors
The enclosed manuscript focuses on the application of dissolvable calcium sulfate beads loaded with differing antibiotic combinations for the localized prevention of osetoarticular infections. The article describes the work as a retrospective study but does detail aspects of the material preparation for interoperative application, primarily noting the combinations of antibiotics being applied to the bead prior to implantation. The discussion for this paper larger relies on comparison with other studies implementing calcium sulfate beads loaded with antibiotic combinations, which also indicate the strong potential of this material for delivery of sustained antibiotic levels to prevent infection. A large challenge for this work is the lack of novel nature given that these materials and the described application as a local delivery source for antibiotics, even in the utilization of antibiotic combinations, has been extensively reported in the literature. Furthermore, as one of the key attributes that are highlighted as an important aspect about the applied calcium sulfate beads is the natural breakdown and resorption of these beads by the body it would be important to highlight this evidence via follow up radiographs that can then be compared with the post-operative images.
Apart from these major concerns, there do appear to be multiple cases in which the terminology used would dramatically change the intended meaning, with most notable of these being the title, "The Use of Dissolvable Synthetic Pure Calcium Impregnated with Antibiotic in Osteoarticular Infection in Patients with Diabetes," which identifies the material being utilized in the study as "Pure Calcium". There are a few instances of this throughout the paper and can create confusion for readers.
As the combination of different antibiotics during treatment appears to be the major highlight within this paper too it seems strange that the focus should be shifted to the calcium sulfate beads in title and introduction. On the whole the paper actually focuses very little on the material aspects of these beads apart from the couple of comparisons with PMMA. As there are many substrates designed for implantable synthetic materials capable of local sustained delivery of antibiotics it would be appropriate to identify other materials for comparison apart from just PMMA if this is the main focus of the paper. If instead the combination of antibiotics is the focus it would be my recommendation to make some alterations to the introduction, discussion, and title to better reflect this.
Based on these points it is my recommendation to reject this article as it appears to need substantial modification to address the core narrative and focus.
Comments on the Quality of English LanguageThere were only minor grammatical errors detected and the major issue was found in the title that describes the utilized material as "pure calcium" which does not accurately portray the substrate being used in the study.
Author Response
Dear Reviewer,
We appreciate your thorough feedback on our manuscript. We have thoroughly evaluated each of your points and will address them respectfully and constructively.
1. Preparation and Application During Surgery
Your concern regarding the insufficient detail in the description of the material preparation and intraoperative application of the calcium sulfate beads has been noted. Upon review, it is evident that this point requires a more comprehensive explanation. This section has been significantly expanded to include detailed information on the methods of antibiotic selection, bead preparation, and setting times, along with an intraoperative image to illustrate the process. This clarification aims to address your concern. The detail provided, as recognized by the other two reviewers, enhances readers' understanding of our methodology.
2. Original Contribution of the Research
The proposal of utilizing calcium sulfate beads for localized antibiotic delivery is not novel, as it has been extensively documented in existing literature. Upon further exploration of the PubMed database, we identified 244 articles pertaining to "calcium sulfate beads," yet only 17 of these specifically address orthopedic applications in diabetic patients. The context of our research—osteomyelitis and osteoarticular infections in diabetic patients—represents a notably underexplored domain. This study addresses a gap in the literature by examining the intersection of calcium sulfate beads and infection management in diabetic orthopedic cases. Considering the increasing population of diabetic patients susceptible to infections and complications after orthopedic procedures, our findings provide valuable contributions and new insights to this area of study. This aspect requires further attention.
3. Radiographic Evidence of Resorption
The necessity for radiographic evidence of bead resorption was highlighted to support our assertions regarding the biodegradability of calcium sulfate. This point is significant, and we have included follow-up radiographs that illustrate the resorption and integration of the beads into new bone in two distinct clinical cases. The images and accompanying descriptions, now incorporated into the manuscript, offer clear visual evidence of the complete resorption of the beads over time and their function in facilitating bone regeneration.
4. Terminology and Title
The terminology employed, especially the reference to "Pure Calcium," may lead to confusion. We acknowledge this feedback and have made revisions to the title and manuscript text accordingly. The title has been revised to: "The Use of Dissolvable Synthetic Calcium Impregnated with Antibiotic in Osteoarticular Infection in Patients with Diabetes." This revision more accurately represents the material and aims to eliminate potential confusion for readers. Your observation regarding the quality of the language is noted. As non-native English speakers, we aim to present our work with clarity; however, we will utilize the MDPI Author Services for additional refinement.
The manuscript's focus is outlined in this section.
The concern was raised regarding the manuscript's shifting focus between calcium sulfate beads and antibiotic combinations, with a suggestion to refine the narrative. Both aspects are essential to the treatment's success; the beads serve as a delivery vehicle, while the antibiotics act as the agents combating infection. We acknowledge your perspective. The introduction and discussion sections have been revised to enhance alignment with the study's focus and clarify its key objectives. Demonstrating the effectiveness of both the beads as a vehicle and the antibiotics as agents is pertinent to the manuscript's contribution to the literature.
We note that the other two reviewers offered highly positive feedback, commending the significance of our findings and the clinical relevance of our work. Their suggestions have been considered, and revisions have been made to enhance the manuscript's clarity and impact.
We appreciate your review and trust that our revisions adequately address your concerns. The modifications implemented address your feedback and enhance the overall quality of the manuscript. I appreciate your time and consideration.
Best regards,
Dr. Mihnea Ioan-Gabriel Popa and the research team
Round 2
Reviewer 3 Report
Comments and Suggestions for Authors
After reviewing the resubmitted material it is clear that the authors have adequately addressed original concerns with the enclosed manuscript and in it's current state the work does stand to as a valuable resource for researchers in this field.